# The Effects of Modified Atmosphere Packaging on the Quality Properties of Water Buffalo Milk’s Concentrated Cream

**DOI:** 10.3390/molecules28031310

**Published:** 2023-01-30

**Authors:** Mubin Koyuncu, Songül Batur

**Affiliations:** 1Food Engineering Department, Faculty of Engineering, Iğdır University, 76100 Iğdır, Turkey; 2Research Center for Redox Applications in Foods, Iğdır University, 76100 Iğdır, Turkey

**Keywords:** cream, hydrogen, lipolysis, modified atmosphere packaging, nitrogen, oxidation, water buffalo milk

## Abstract

Concentrated cream (CC) is a dairy product containing more than 60% milk fat. CC has a very short shelf life because it is made from unripe cream. The present study aims to determine how packaging with reducing gas (H_2_) and nitrogen (N_2_) affects the quality properties and shelf life of CC. For this purpose, lipolysis, oxidation, color, microbiological, and free fatty acid development and the fatty acid composition of modified atmosphere packaged (MAP) CC samples were studied for 28 days. For MAP1, 96% N_2_ + 4% H_2_ was used, and for MAP2, 100% N_2_ and air was used for the control group. During storage, MAP1 samples remained at lower lipolysis (ADV and FFA) and oxidation levels than MAP2 and the control group. The MAP1 and MAP2 methods preserved the color of the samples and reduced the microbial growth rate. A lower formation of free fatty acids was observed in the samples packed with MAP1 and MAP2 than in the control group. The results showed that hydrogen gas positively affected the quality and storage time of samples.

## 1. Introduction

Food manufacturers and researchers face one of their biggest challenges when trying to explore how to extend the shelf life of food. The technology known as modified atmosphere packaging, or MAP, is often used as a means to achieve this goal. However, the spoilage reactions (hydrolysis, oxidative reaction, and microbial growth) that can occur in foods during the storage period result in a loss of nutritional value, sensory properties, and economic value of the product, which limits the possible applications of the MAP method. The use of the modified atmosphere packaging method, in which a reducing gas such as hydrogen is introduced, makes it possible to circumvent these limitations [1,2]. H_2_ gas has many characteristic properties that make it unique and suitable for certain applications. H_2_ is the lightest gas (14 times lighter than air) and can penetrate the core of biological systems in molecular form. Only highly reactive free radicals are negatively affected by hydrogen [3]. Hydrogen is highly flammable at a range of 4–75% (*v*/*v*) in the air, and explodes at 18.3–59% (*v*/*v*). By the dilution of hydrogen with nitrogen lowers the risks [4].

The advantage of MAP with H_2_ gas is that it has an effect that slows down the growth of microorganisms, and prevents the formation of free radicals, damage to vitamins, such as C and E and beta-carotene, the degradation of pigments, such as carotenoids and rancidity products, that cause loss of flavor and spoilage of food [2]. It is thought that these benefits of the method will provide an advantage in packaging foods with high-fat content. This method describes a new application technique of modified atmosphere packaging with nitrogen and hydrogen for foods with high-fat content. The quality of a variety of dairy products, including fresh cheese [2] and dairy beverages [5], is maintained by the use of hydrogen gas.

Water buffalo milk is a nutritious alternative to conventional cow’s milk because it contains higher fat, protein, lactose, and mineral content than cow’s milk [6,7]. The fat of buffalo milk contains more saturated fatty acids, including palmitic acid, trans fatty acids, linolenic acid (ω3), and conjugated linolenic acid [7]. The fat globules of water buffalo milk have a diameter of 2.1 to 4.0 mm (average 2.80 mm) and facilitate the natural creaming of buffalo milk [8], thus making it more suitable for CC production. The correlation between the size of the fat globules and the percentage of unsaturated fatty acids suggests that water buffalo milk butter has a longer shelf life than its bovine milk counterpart due to its higher percentage of solid fat. A higher percentage of solid fat would result in slower fat hydrolysis and less rancidity [8]. Concentrated cream (CC) is a cream that contains at least 60 per cent milk fat. CC is produced in many Asian and Balkan countries under names, such as kaymak, kajmak, kaimak, or gemagh [9]. Although CC made from water buffalo milk tends to be preferred, other types of milk, such as cow’s, sheep’s and goat’s milk, can be used for its production [9,10]. In the present study, we also preferred water buffalo milk for the preparation of CC.

Fatty acids (FA) are responsible for many of the processes necessary for normal physiological functioning and maintenance of health. Saturated fatty acids, also known as SFAs, perform an important role in energy production, energy storage, lipid transport, synthesis of phospholipids, and sphingolipids needed for membrane formation, and covalent modification of a variety of regulatory proteins. Monounsaturated fatty acids (MUFA) are also involved in many of these processes and perform an important role in maintaining the optimal fluidity of the lipid bilayer of the membrane. The term “essential” is used for certain polyunsaturated fatty acids (PUFA) that cannot be synthesized in the human body and are essential for health [11,12]. Free fatty acids (FFA) originate from lipolysis in foods. Caballero et al. reported that the degree of lipolysis in fats and oils can be studied by determining the FFA profiles [13].

This study aims to determine the influence of MAP with H_2_ gas (MAP1) on lipolysis, oxidation, microbiological growth, FFA production, and FA profiles in CC samples of water buffalo milk. It also aims to investigate the possibilities of using this method for the preservation of fatty foods.

## 2. Results and Discussion

### 2.1. Raw Buffalo Milk’s Composition

The fat, solid non-fat, and protein contents (Table 1) of the analyzed milk are compatible with the literature [14,15].

### 2.2. Detection of Residual Gasses in the Packaging Atmosphere

Gas measurements taken after 28 days in the headspace of the packages are presented in Table 2. At the end of the storage period, the hydrogen content of the samples packaged with MAP1 decreased from 4% to 1.6%. It has been reported that the 0.1% hydrogen remaining in the package during storage still maintains its effectiveness as a reducing agent [1]. Depending on the permeability of the packaging material used in MAP processes, gas losses may occur in the package. In a MAP study, in which hydrogen was added to the gas mixture, the presence of 0.04% hydrogen was detected in the gas content of the package at week 12 [16]. A small amount of oxygen, which was thought to be trapped in the structure of the cream during the application, was found in the MAP-treated samples.

### 2.3. Microbiological Analysis

The changes in the number of yeast molds and total aerobic mesophilic bacteria (TAMB) of the samples during the storage period are shown in Table 3. The number of yeast molds in the control group increased significantly from the beginning of the storage period, and this growth was significantly different from the number of yeast molds in the MAP1 and MAP2 samples. Although the MAP1 and MAP2 samples had lower numbers of yeast molds than the control group during the storage period, there was no significant difference between the sample groups after day 21. It is probably due to the developing acidity in control samples slowed down the microbial growth. The increase in acidity and biochemical components that occur during the ripening of foods inhibit the growth of microorganisms [17]. During the storage period, the TAMB count increased faster in the control group than in the MAP1 and MAP2 samples, resulting in significant differences.

Yeast–Mold counts of Afyon Kaymak (a concentrated cream product made exclusively from water buffalo milk) samples collected from the market were determined in the range of 3.78 to 6.48 log CFU/g, and TAMB counts were in the range of 4.15 to 6.81 log CFU/g [18]. It has been reported that the average number of yeast–molds of concentrated creams collected from the market and produced by traditional methods is 3.32 log CFU/g, and the average number of TAMB is 6.18 log CFU/g [19].

The modifications in the air pressure inside the package caused by MAP affect the metabolism of the product and slow down the growth of microorganisms, which prolongs the shelf life of the product and maintains its quality and appearance [20]. By restricting O_2_, undesirable chemical and enzymatic reactions and microbiological development are inhibited in MAP systems. By lowering the oxido-reduction potential of the atmosphere to negative values using a reducing agent, such as hydrogen gas, oxidation reactions are restricted because the abundance of free electrons available for biochemical reactions effectively neutralizes oxidative compounds, such as the extremely harmful hydroxyl radical. The addition of molecular hydrogen selectively reduces oxygen radicals, limiting lipid oxidation (rancidity) in cells and the resulting proliferation of aerobic bacteria [21]. Many studies report that MAP applications inhibit microbial growth [20,22]. Additionally, a study reports that MAP with hydrogen application inhibits microbial growth [2].

### 2.4. Biochemical Analysis

When microorganisms grow in food, there are some changes in the biochemical structure of the food, such as the changes that occur during the fermentation process as a result of the metabolic activities of the microorganisms [23]. With the production of organic acids, the acidity increases and the pH decreases during fermentation [24]. In addition, yeast and molds that develop on food have proteolytic and lipolytic effects [25]. Considering the above data and the number of TAMB and yeast molds that developed during the storage period of the samples, the acidic and lipolytic development in the control samples is due to microbial growth. The microflora of whole milk tends to be present in the cream component. Microorganisms associated with the lipid hydrolysis of triglycerides to free fatty acids can produce increased acidity, rancidity, and soapiness [26].

The results of the biochemical analysis of the samples are shown in Table 4. The titratable acidity, lipolysis, and peroxide values of the samples increased during storage. The titration acidity values of MAP1 samples packaged with an H_2_ gas mixture were lower than those of MAP2 and the control group. The acidity values of the control group increased significantly from the 14th day of storage. The acidity of the cream should not exceed 0.225% as far as the percentage of lactic acid is concerned [27]. The acidity of the MAP1 samples remained below the limits during the storage period. The MAP2 samples remained below the limits for 21 days.

When the acidity values of MAP-packaged traditional concentrated cream (Afyon Kaymak, produced from buffalo milk) samples were examined, it was seen that CO_2_ and N_2_ gases were used at different rates in their packaging, and that all samples reached higher acidity levels than the MAP1 sample values of our study [28]. In another fresh Afyon Kaymak samples study, acidity levels were measured close to our initial sample’s acidity level [29]. The fact that the average acidity values of fresh, concentrated creams produced by traditional methods exceed the upper limits determined by the authorities [27] has been associated with high-temperature (10 °C) storage [19]. In another study performed on traditional concentrated cream collected from the market, the average acidity values of the samples were determined at the level of 0.15 ± 0.05 [9]. It has been reported that the acidity values of butter washed with hydrogen-rich water have significantly lower values than butter washed with normal water after 90 days of storage [30].

On the other hand, the acidity value of the control samples on the 14th day was close to the upper limit of acidity determined for the cream. At the end of the 21st and 28th days, the lipolysis values of the MAP1 and MAP2 samples were significantly lower than those of the control group. At the end of the storage period, the highest lipolysis value in samples was 0.362 in the control sample. The lowest lipolysis value of 0.264 was measured in the MAP1 sample. It has been reported that after 90 days of storage, butter washed with hydrogen-rich water undergoes slower lipolysis and oxidation than butter washed with normal water [30].

The peroxide values of the control samples increased rapidly and caused significant differences. The peroxide values of the MAP1 and MAP2 samples increased rapidly on day 21. At the end of storage, the peroxide values began to decrease as secondary products are formed from the products of the first oxidation during the second phase of lipid oxidation [31].

Due to slower microbial growth, especially during the initial storage period in the MAP1 samples, acidic and lipolytic development were also slower. MAP recommends 100% nitrogen for many foods. Nitrogen is an inert gas used to replace air in oxygen sensitive packaged products, and is also used to inhibit the growth of aerobic microorganisms [26]. However, nitrogen and the reducing effect of hydrogen seem to slow the growth of microorganisms in the package even more.

Since day 14, the peroxide levels of the MAP1 samples had remained at the lowest level, which can be attributed to the effect of H_2_ on controlling oxidation in the MAP1 samples. This scenario can be attributed to the reducing abilities of H_2_ on oxygen and radicals in the packaging and product [32]. Khan et al. determined the peroxide values of raw, pasteurized, and boiled buffalo milk on the 6th day of storage (4°C) at 0.31 ± 0.06, 0.37 ± 0.10, and 0.51 ± 0.07, respectively [14].

### 2.5. Color Properties

The light, process temperature, pH, or oxygen capacity can change the color pigments of foods [33]. Additionally, spoilage by microorganism especially by yeast and molds can cause to surface discoloration [26]. Researchers attributed increase in b* value problem that occur during storage to the non-enzymatic degradation of milk (Maillard reaction) [34]. The values and range of the color characteristics (L*, a*, b*, and ΔE) of the cream samples are shown in Figure 1. The data of the samples and the standard deviations are in Appendix A (given in Appendix A). When examining the L* values, it was found that the control samples lose brightness over time, the brightness of the MAP1 samples is very close to the values of the first day, while the MAP2 samples maintain their brightness on the first day. The a* values of the MAP1 and MAP2 samples remained close to the first-day values during the storage period. It was observed that the a* values of the control samples changed from green to red. Over time, the b* values of the samples became increasingly yellow. While this change was limited for the MAP1 and MAP2 samples, the control samples moved rapidly toward yellow. Looking at the overall color change (ΔE), it was observed that the MAP1 and MAP2 applications kept the color of the samples close to the values of the first day. It was also observed that the color values of the control samples differed from the values of the first day from the 14th day onwards.

Changes in physicochemical processes are reflected in the color of dairy products. The most important indicator of quality is the color change that occurs during the storage period [35]. Compared to the control samples, the MAP1 and MAP2 samples were less affected by the increasing number of microorganisms and the biochemical reactions, so that their colors remained close to the initial values.

### 2.6. Chromatographic Analyzes

#### 2.6.1. Fatty Acid Composition of CC Samples

The amount of fatty acids contained in milk depends on several factors, including animal species, breed, stage of lactation, daily routine, quality and quantity of feed consumed by the animal, geographical location, and season [36,37]. Appendix A (given in Appendix A) shows the FA content of the samples. The most abundant fatty acids in the samples are palmitic acid (C16), oleic acid (C18.1 C), and myristic acid (C14). Many studies on water buffalo milk and its products report similar results [6,7,8,14]. The fatty acid chromatograms of the samples from the 28th day of storage, and the most common fatty acids are shown in Figure 2.

Some SFAs (C4, C8, and C10), MUFAs (except C18.1T), and some PUFAs (C18.3, C18.3 N6, and C20.4) were significantly affected by the applications. The majority of fatty acids in the MAP1 and MAP2 packaging samples had higher concentrations than in the control group. The N3 and N6 fatty acids cannot be converted into each other, and both are essential nutrients [38]. In the present study, a remarkable content of PUFA was found in the samples. Rustan and Drevon (2001) found that fatty acids can be converted into each other (N3 to N3, and N6 to N6). According to our study, this conversion between PUFAs occurred at the end of the storage period.

The SFAs in the MAP1, MAP2, and control samples at day 28 were 69.603 ± 0.394%, 69.543 ± 0.354%, and 70.290 ± 0.187%, respectively. One study reported that dairy products from buffaloes were analyzed, and 65.30% SFA was found in samples of water buffalo milk product kajmak (high-fat cream product), and 70.82% SFA in water buffalo butter in the same study, MUFA and PUFA in kajmak samples were 30.52% and 4.18%, respectively, while MUFA and PUFA in butter samples were 26.28% and 2.09%, respectively. In our study, MUFA and PUFA at the end of storage were 23.789 ± 0.027% and 1.792 ± 0.010% in MAP1 samples; 24.209 ± 0.139% and 1.772 ± 0.030% in MAP2 samples; and 23.654 ± 0.225% and 1.756 ± 0.012% in control samples, respectively. Romano et al. (2011) analyzed the fatty acids of Mozarella di Bufala Campana cheese samples produced in different seasons and found that the SFA, MUFA, and PUFA values in cheese samples from winter were 66.31%, 27.50%, and 4.37%, respectively.

#### 2.6.2. Free Fatty Acids from CC Samples

FFAs are formed in dairy products mainly by enzymatic degradation of glycerides by lipase activity from various sources. FFAs have a low taste threshold and contribute to the taste and flavor of many dairy products, especially fermented dairy products. Accordingly, FFA content and lipase activity can be considered a valuable index of good quality and correct storage of food [39]. The development of lipolysis in the samples, prepared from buffalo milk packaged with MAP1 and MAP2, slowed down considerably at the end of the storage period compared with the control group. At the end of the 28th day, a significant development was observed for all FFAs in the control group. At the same time, FFAs showed less improvement in MAP1 and MAP2 samples (*p* < 0.05), which was statistically significant (Appendix A). At the beginning of storage, none of the control samples had short-chain (C4 and C6), C8, or some long-chain (C15, C16:1, or C18:2) FFAs, but at the end of storage, these FFAs developed in all control samples. The rancidity of dairy products is caused by high concentrations of short-chain FFAs, especially C4:0 [39]. No short-chain FFAs were formed in samples packaged with MAP2. Statistically insignificant amounts of short-chain FFAs occurred in the samples packaged with MAP1. From this point of view, rancidity, which may occur in the samples at the end of 28 days, was prevented by the application of MAP1 and MAP2. This can be accounted for by the fact that the lipase activity originating from the microorganism could not develop when MAP1 and MAP2 were applied. In Section 3.2, it was mentioned that the yeast mold and TAMB counts of the samples treated with MAP1 and MAP2 are lower than those of the control samples, which affects the biochemical properties, such as lipolysis.

In a study that examined 14 samples of Afyon Kaymak (a concentrated cream product made exclusively from water buffalo milk) [29], the FFA values were higher than all FFAs in our fresh samples. The FFA values in the present study were lower than the values in the control samples obtained at the end of day 28. Application of MAP1 and MAP2 resulted in lower FFA values at the end of day 28 than most values in this study. It is noteworthy that after 28 days, short-chain FFAs were lower in MAP1 and MAP2-treated samples than in fresh Afyon Kaymak samples. Similar results were found in a study with cream butter made from sheep and cow milk [40]. The FFA values for the control group samples on day 28 of our study were higher than the FFA values found in the literature [40]. The FFA values of samples packaged with MAP1 and MAP2 were lower than the FFA values of cream butter samples prepared from sheep and cow milk (except for C18 and C18.1 bovine cream butter FFA). Slowing of free fatty acid development (inhibition of lipolytic degradation) of butter washed with hydrogenated water may be related to the reducing property of hydrogen [30]. GC-MS chromatography of Free fatty acids of samples from day 28 are shown in Figure 3.

The MAP with hydrogen technique had significant effects on the microbiological, biochemical, and chromatographic properties of cream samples. With this technique, it is assumed that the reducing property of hydrogen has a suppressive effect on microbial growth, therefore microbial acidity development and lipolytic and oxidative reactions proceed more slowly. This leads to improved physical and chemical stability of cream, increased shelf-life and safety, and better color properties. So, this technique had a profound impact on the quality of the cream samples tested.

## 3. Materials and Methods

### 3.1. Preparation of CC Samples

Buffalo milk was obtained from local markets in Iğdır Turkey. A 50-L buffalo milk was obtained from a mixture of the milk produced by 11 water buffaloes. The composition of raw milk was determined using a Milk Analyzer (Master Pro, Milkotester, Belovo, Bulgaria) [14].

The CC samples were prepared in January 2021 in the laboratory of the Department of Food Engineering at Iğdır University. The raw milk was filtered, heated to 40 ± 2 °C, and passed through a separator to obtain the cream. A manual household cream separator was used for separation. The cream was analyzed for fat content by the Gerber method [41], and 68% *w*/*w* fat was detected. The cream was pasteurized at 85 °C for 30 min and cooled down to 8–10 °C for 2 h. The cream was then divided into 50 g portions in a PET cups. The sample cups were stored at 4 °C until the packaging procedure.

Product packaging and MAP1 and MAP2 applications were performed in the laboratory of Redox Applications at the Food Research Center, Iğdır University. Different gas mixtures of nitrogen (N_2_) and hydrogen (H_2_) were prepared using a gas mixer (Accuracy ± 2%) (Dansensor, MAP Mix 9001 ME, Ringsted Denmark). The CC samples were packaged with a modified atmosphere packing machine (Lipovak, KV -600, Sakarya, Turkey) using polyethylene laminated polystyrene containers and polyethylene films (Çokay Plastik, Sakarya, Turkey). Three groups of CC were prepared. The first group was packaged with MAP1, the second with MAP2, and the third with air (control). The gas content of MAP1 packaging was 96% N_2_ + 4% H_2_. The gas content of MAP2 packaging was 100% N_2_. The samples were stored at 4 °C for 28 days. High-purity gasses were used for packaging (AnkaraGaz, Ankara-Türkiye). The lower hydrogen ratio could not be used due to the sensitivity range of the gas mixer.

In-package gas ratios were determined using the In-Package Gas Meter (A. Kruess Optronic, Hamburg, Germany). A sealed rubber label was stuck on the package and the device probe was dipped into the package from this point, and the gasses in the ambient atmosphere and their amounts were determined.

### 3.2. Microbiological Analysis

CC samples (10 g) were diluted with 90 mL Ringer’s solution. Total aerobic mesophilic microorganisms and yeast molds were counted in triplicate on PCA (Merck, Darmstadt, Germany) at 30 °C for 2 days and on potato dextrose agar (PDA, Merck, Darmstadt Germany) at 25 °C for 7 days, respectively. Colonies were counted and results are expressed as the logarithm of colony-forming units per gram (log CFU/g) [42].

### 3.3. Biochemical Analysis

CC samples were analyzed for titratable acidity as lactic acid (LA %) [43]. The lipolysis was determined as acidity value (ADV) by the BDI method [44]. An amount of 5 g of the sample was placed in butyrometers and 20 mL of BDI solution (prepared by dissolving 30 g of Triton X-100 and 70 g of sodium tetraphosphate in 1 L of distilled water) was added. The butyrometers were placed in boiling water and the oil was completely released. The samples were centrifuged again for 1 min and kept in a water bath at 57 °C for 5 min. A sufficient amount (1.5–2 g) of the oil collected in the upper part was taken, and 5 mL of oil solvent (petroleum ether/n-propanol; 4:1 *v*/*v*) containing 0.1 g/L thymol blue indicator was added, and titrated with 0.01 N tetra-n-butylammonium hydroxide.

The analyzes for the determination of the peroxide value were performed by spectrophotometry according to the AOAC standard technique [45]. For peroxide analysis, 0.3 g of the oil sample was removed and transferred to a 20 mL glass tube. An amount of 9.6 mL of chloroform/methanol (70:30 *v*/*v*) was added and the oil was dissolved by vortex mixing. Then, 0.05 mL of a 30% ammonium thiocyanate solution was added. Then, 0.05 mL of a ferric chloride solution (0.35% solution containing 2% 10 M HCl) was added. The blank was prepared using the same oil-free solutions as above, and the absorbance values of all tubes were determined by measuring at 500 nm. The concentration was determined from the curve prepared from the absorbance values of ferric chloride in a series (0.25–10 mg/L). The obtained peroxide values are expressed as meq O_2_/kg oil.

### 3.4. Color Properties

A colorimeter (Konica Minolta CR 410, Osaka, Japan) was used to analyze color properties. The values given by the colorimeter were: L*: brightness, a*: Red (+) and Green (−), b*: Yellow (+) and Blue (−), and ΔE: the total color change.

### 3.5. Chromatographic Analyzes

#### 3.5.1. Analysis of the Fatty Acid Composition

The modified method of Ocak et al. [46] was used to determine the composition of FA. CC Samples were melted at 40 °C, and centrifuged at 1200 rpm for 1 min. Approximately 0.2 g of oil was transferred to a tube and dissolved in 2 mL of upper-phase hexane. The methyl esters of the fatty acids were obtained by adding 0.2 mL of a 1 N solution of KOH in methanol and vigorously shaking the mixture. The methyl esters were separated in an Agilent gas chromatograph (7820 A) equipped with a “flame ionization detector” and a capillary column (Restek Rt-2560, 100 m, 0.25 mm, 0.20 µm). Fatty acids were identified by FAME 37 Mix (SUPELCO, Bellefonte, PA, USA) internal standards.

#### 3.5.2. Analysis of Free Fatty Acids

FFAs were determined by gas chromatography-mass spectroscopy (Thermo Fisher Trace ISQ GC-MS, Waltham, MA, USA). Aminopropyl SPE columns (Agilent Technologies, Santa Clara, CA, USA) were used for the separation of FFAs using the modified methods of De Jong and Badings [47].

Two grams of sample CC were used for analysis, and 0.5 mg of pentanoic acid and tridecanoic acid (Sigma-Aldrich, Milwaukee, WI, USA) were used as internal standards. The peak area of pentanoic acid was used for the calculation of short-chain FFAs and the peak area of tridecanoic acid was used for the medium- and long-chain FFAs [48]. FFAs were separated using the TRB-FFAP capillary column (60 m 0.25 mm 0.25 m; Teknokroma, Barcelona, Spain). The mass spectral libraries of Wiley9 and Mainlib were used to compare the fatty acids and find out which ones they were [49].

### 3.6. Statistical Analysis

All data were statistically analyzed using SPSS 20 statistical software (SPSS Inc., Chicago, IL, USA). Analysis of variance (ANOVA) and Tukey’s multiple range tests were used to determine whether or not there were significant differences in the results.

## 4. Conclusions

In this study, the effects of modified atmosphere packaging with hydrogen and nitrogen on the shelf life of creams were compared with those of air packaging (control group). At the beginning and the first half of the storage period, MAP1 samples have lower yeast–mold and total aerobic mesophilic bacteria counts. The acidity of the MAP1 samples remained below the standard limits for 28 days. Lipolysis levels in the MAP1 samples showed the lowest development of lipolysis among the samples, although they slightly increased at day 28. Similarly, the peroxide values of the MAP1 samples also increased less than the other samples. The color characteristics of the MAP1 samples were preserved throughout the storage period. Considering the chromatographic properties, no significant change was observed in the fatty acid composition of the MAP1 samples. However, it was determined that the formation of free fatty acids resulting from rancidity was slowed down with MAP1.

Yeast–Mold and total aerobic mesophilic bacteria counts of MAP2 samples increased during the storage period. However, the lowest microorganism counts were counted in MAP2 samples in the second half of storage. Although the acidity, lipolysis, and peroxide values of MAP2 samples were higher than MAP1 samples, it was determined that they were significantly lower than control samples. The color characteristics of the MAP2 samples were preserved during storage and remained close to their first-day values. With the MAP2 application, the lowest FFA development among the samples was determined.

By the controlling the yeast–mold counts, total aerobic mesophilic bacteria count, and acidic, lipolytic, and FFA development the hydrogen gas incorporation in modified atmosphere packaging was positively affected the shelf life of the samples. Considering the results obtained, the shelf life of the cream samples could be extended up to 28 days by the method applied (MAP with the hydrogen). Modified atmosphere packaging with hydrogen has potential for use in fatty foods. However, further investigation with different gas combinations is required to confirm the findings.

## Figures and Tables

**Figure 1 molecules-28-01310-f001:**
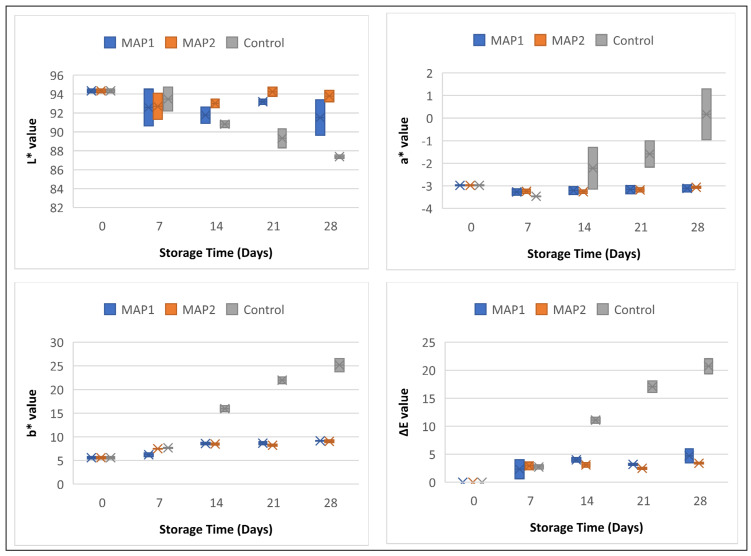
Color characteristics (L*, a*, b*, and ΔE) of CC samples. MAP1: %96 N_2_ + %4 H_2_; MAP2: %100 N_2_.

**Figure 2 molecules-28-01310-f002:**
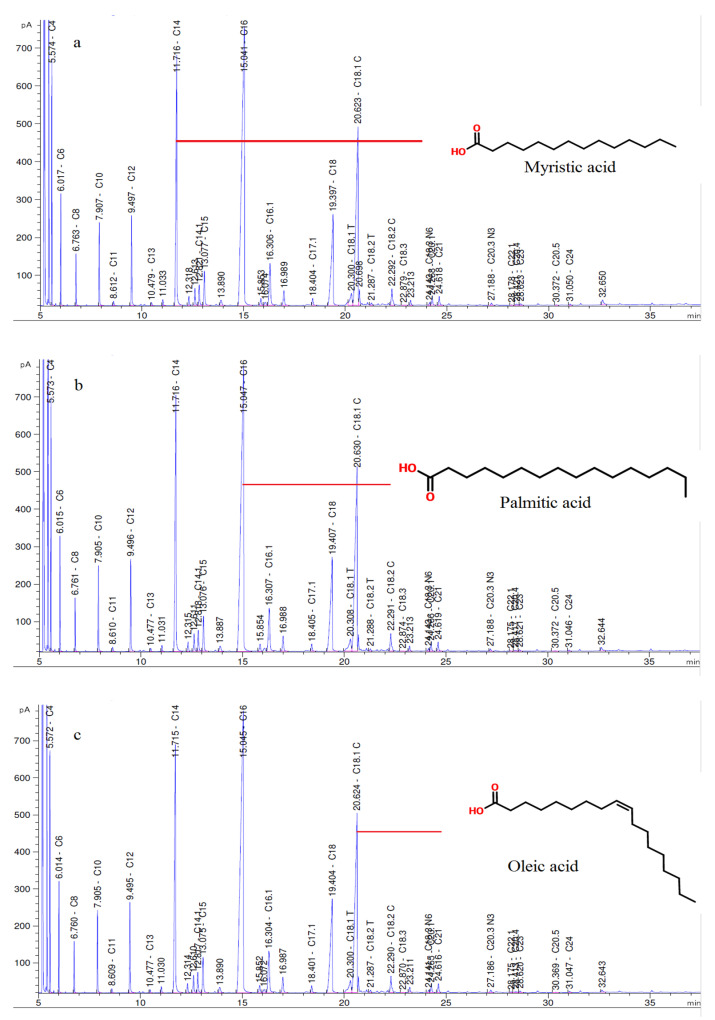
Fatty acid chromatograms of samples (**a**) MAP1, (**b**) MAP2, and (**c**) Control in day 28, and most abundant fatty acids (Miristic (C14), Palmitic (C16) and Oleic (C18.1) acids) of samples.

**Figure 3 molecules-28-01310-f003:**
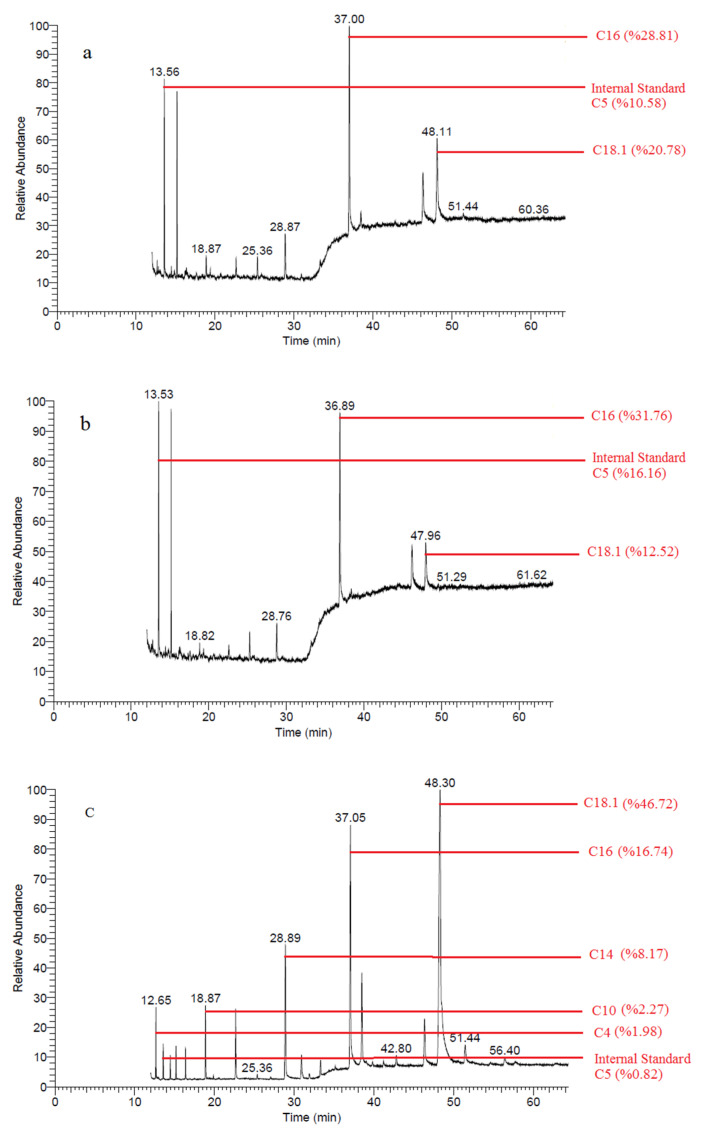
Free fatty acid chromatograms of samples (**a**) MAP1, (**b**) MAP2, and (**c**) Control in day 28.

**Table 1 molecules-28-01310-t001:** Chemical composition of raw buffalo milk (%).

Parameter	%
Fat	6.9 ± 0.14
Solid non-fat	10.45 ± 0.21
Protein	3.75 ± 0.07

**Table 2 molecules-28-01310-t002:** Gas rates in sample packs after 28 days.

	O_2_	N_2_	H_2_
MAP1 ^1^	%0.10	%98.10	%1.60
MAP2 ^2^	%0.10	%99.30	20 ppm
Control	%4.70	%82.30	0

^1^ MAP1: 96% N_2_ + 4% H_2_. ^2^ MAP2: %100 N_2_.

**Table 3 molecules-28-01310-t003:** Yeast–Mold counts, and total aerobic mesophilic bacteria (TAMB) counts of CC samples.

Storage Time (Day)					
	Initial sample *	7	14	21	28
Yeast–Mold Count (log CFU/g)					
MAP1 ^1^	1.58 ± 0.10 ^Ac^	2.27 ± 0.07 ^Cc^	4.61 ± 0.01 ^Bb^	7.23 ± 0.31 ^Aa^	7.51 ± 0.21 ^Aa^
MAP2 ^2^	1.58 ± 0.10 ^Ad^	3.55 ± 0.32 ^Bc^	5.30 ± 0.01 ^Bb^	6.69 ± 0.43 ^Aa^	7.49 ± 0.65 ^Aa^
Control	1.58 ± 0.10 ^Ad^	4.91 ± 0.02 ^Ac^	6.76 ± 0.57 ^Ab^	8.02 ± 0.66 ^Aab^	8.95 ± 0.19 ^Aa^
TAMB Count (log CFU/g)					
MAP1	1.56 ± 0.16 ^Ac^	2.31 ± 0.04 ^Cc^	5.20 ± 0.14 ^Bb^	6.68 ± 0.27 ^Aa^	7.03 ± 0.53 ^Ba^
MAP2	1.56 ± 0.16 ^Ae^	3.42 ± 0.16 ^Bd^	5.44 ± 0.02 ^Bc^	6.31 ± 0.17 ^Ab^	6.95 ± 0.13 ^Ba^
Control	1.56 ± 0.16 ^Ad^	4.67 ± 0.11 ^Ac^	6.45 ± 0.25 ^Ab^	7.82 ± 0.59 ^Aa^	8.45 ± 0.30 ^Aa^

* The initial sample was analyzed before each application. ^1^ MAP1: 96% N_2_ + 4% H_2_. ^2^ MAP2: %100 N_2_. In each application, different superscript capital letters indicate differences between the samples (*p* < 0.05). Different superscript lowercase letters indicate differences between storage times (*p* < 0.05).

**Table 4 molecules-28-01310-t004:** The biochemical analysis results of CC samples.

Storage Time (Day)					
	Initial sample *	7	14	21	28
Acidity (%LA)					
MAP1 ^1^	0.11 ± 0.01 ^Ab^	0.12 ± 0.00 ^Ab^	0.17 ± 0.00 ^Ca^	0.18 ± 0.02 ^Ba^	0.20 ± 0.01 ^Ca^
MAP2 ^2^	0.11 ± 0.01 ^Ac^	0.12 ± 0.01 ^Ac^	0.19 ± 0.05 ^Bb^	0.21 ± 0.01 ^Bb^	0.28 ± 0.29 ^Ba^
Control	0.11 ± 0.01 ^Ad^	0.11 ± 0.00 ^Ad^	0.21 ± 0.01 ^Ac^	0.67 ± 0.02 ^Ab^	0.89 ± 0.03 ^Aa^
Lipolyses (ADV)					
MAP1	0.10 ± 0.00 ^Ab^	0.10 ± 0.00 ^Ab^	0.14 ± 0.02 ^Ab^	0.14 ± 0.00 ^Bb^	0.26 ± 0.03 ^Ba^
MAP2	0.10 ± 0.00 ^Ac^	0.11 ± 0.01 ^Abc^	0.13 ± 0.02 ^Abc^	0.15 ± 0.03 ^ABb^	0.27 ± 0.00 ^Ba^
Control	0.10 ± 0.00 ^Ac^	0.12 ± 0.01 ^Abc^	0.17 ± 0.02 ^Abc^	0.19 ± 0.01 ^Ab^	0.36 ± 0.04 ^Aa^
Peroxide Value (meq O_2_/kg)					
MAP1	0.05 ± 0.00 ^Ab^	0.19 ± 0.01 ^ABab^	0.11 ± 0.04 ^Bab^	0.38 ± 0.18 ^Aa^	0.35 ± 0.22 ^Aa^
MAP2	0.05 ± 0.00 ^Ab^	0.10 ± 0.06 ^Bbc^	0.18 ± 0.04 ^Bbc^	0.46 ± 0.24 ^Aa^	0.44 ± 0.16 ^Aa^
Control	0.05 ± 0.00 ^Ad^	0.24 ± 0.04 ^Ac^	0.47 ± 0.08 ^Ab^	0.76 ± 0.10 ^Aa^	0.48 ± 0.02 ^Ab^

* The initial sample was analyzed before each application. ^1^ MAP1: 96% N_2_ + 4% H_2_. ^2^ MAP2: %100 N_2_. For each application, different superscript capital letters show the differences between the samples (*p* < 0.05). Different superscript lowercase letters indicate differences between storage times (*p* < 0.05).

## Data Availability

Not applicable.

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
