# Peer review of "The Effects of Modified Atmosphere Packaging on the Quality Properties of Water Buffalo Milk’s Concentrated Cream"

_molecules, 2023, doi:10.3390/molecules28031310_

Round 1

Reviewer 1 Report (Previous Reviewer 2)

Introduction - to be completed, there are no specific comparisons from the literature, Results - what statistical hypothesis, there is no theoretical and numerical model, no discussion, conclusions should be grouped and supplemented with all the results.

Author Response

Dear Reviewer
Thank you again for the time you spent on our work.
In this experimental study we conducted with my graduate student under very difficult conditions during the COVID-19 pandemic, we investigated the effects of MAP application on cream samples. We tried to convey the results we obtained in the study. We did not do modeling work. We have explained the purpose of the study, studies conducted and published with the same application, analysis results, and literature comparisons in the article. We think that we have made all the necessary revisions to our work. We hope for your positive comments.

Reviewer 2 Report (Previous Reviewer 1)

The manuscript has been resubmitted with minor revision. The improvements are far from satisfactory. Nonstandard expressions is still full of all the manuscript. The defects pointed out previously still have not been satisfactorily handled. Considering the authors' persistent support for the journal, one more major revision is given.

1.Abstract. 

The authors have been given several chances to improve the manuscript, but the abstract part is still not satisfactory. Even the abstract part is not a major part in the first review, the author should try to improve all the manuscript. In the abstract, the first two sentences introduce CC, which is a typical "introduction" content. In the middle, a lot of content was devoted to introducing the used methods. At last, it is general results. It lacks specific targeted data and conclusions.

2.Introduction.

L45: than conventional cow's milk.

The authors insist to use lots of content to introduce water buffalo milk in the introduction, but it still has little relationship with the theme of the study. Maybe the authors can change their title to “The effects of modified atmosphere packaging with nitrogen and hydrogen on the quality properties of water buffalo milk’s concentrated cream” and do some modification throughout all the manuscript.

3.Materials and Methods

L77-78: Please show more details of the method to measure milk composition.

L81: Table 1 should be the experimental results. What does the data in this table mean? Mass percentage ? Or concentration ? Why is there no unit ? Why are the decimal places inconsistent? Why use , instead of a decimal point ?

L85: All used instruments should be provided with models, suppliers and addresses.

L87: How to get 68 % here. If it is mass percentage, please use “” % w/w.

L100: 4 °C for 28 days

L108: CC samples (10 g) were

L108: give more details and reference about Ringer's solution.

L116: 20 mL

L117: 1 L

L120: 5 mL

L125: 20 mL

L127: 0.05 mL

L134: A colorimeter (CR 410, Konica Minolta Co, Osaka, Japan) was. Please modify all used instruments according to the format of the journal.

L139: Ocak et al. [19]

L141, 142:  L

L150-151: De and Badings [20]. The authors should standardize the use of references according to the format required by the journal.

4. Results and Discussion

L171-172: Incomplete sentence.

L171-172: Not a good explanation.

L177: Table 2: nonstandard data representation.

L205: Table 3: nonstandard data representation.

L231: Table 3: nonstandard data representation.

L275: Figure 1: nonstandard representation

It still lacks in-depth analysis and comparison with similar studies.

5.Conclusions

Too much content is the result rather than the conclusion.

Author Response

Dear Reviewer
Thank you again for the time you spent on our work.
In this experimental study we conducted with my graduate student under very difficult conditions during the COVID-19 pandemic, we investigated the effects of MAP application on cream samples. We tried to convey the results we obtained in the study. We have explained the purpose of the study, studies conducted and published with the same application, analysis results, and literature comparisons in the article. We think that we have made all the necessary revisions to our work. We hope for your positive comments.

Round 2

Reviewer 1 Report (Previous Reviewer 2)

Please group and complete the discussion separately.

Reviewer 2 Report (Previous Reviewer 1)

It is the fourth time to review this manuscript. Although some revisions of the manuscript is still not completely satisfactory, it has addressed almost all comments.  It may meet the publication requirements of the journal.

This manuscript is a resubmission of an earlier submission. The following is a list of the peer review reports and author responses from that submission.

Round 1

Reviewer 1 Report

Introduction

1. “This method is a new application technique used in high-fat foods for the first time.” Even different researchers may have different viewpoints for the definitions of high-fat food, it is obviously not the first application of similar MAP on these foods.

2. “Water buffalo milk is more nutritious” In some cases, there are significant differences in nutrient contents for similar food substances. However, from a rigorous academic point, it can hardly to prove that one food is “more nutritious” than its similar competitors.

The authors used this paragraph to explain that water buffalo milk is “more nutritious” than cows milk (even water buffalo milk also belongs to cows milk), but actually it has little relationship with the theme of the study.

3. “The ratio of unsaturated fatty acids has a positive correlation with fat globule size, suggesting that buffalo milk fat will have better retention than its bovine counterpart due to higher unsaturated fat content.” It is difficult to understand why it can suggest, while there is no reference here.

4. Water buffalo milk was preferred in CC production. ” The content and references from the manuscript are not enough to support this conclusion.

Materials and Methods

5. “CC samples were produced in the IÄŸdır University Food Engineering Department laboratory in January 2021.” More details about instruments and processes of CC preparation should be exhibited .

6. with 68% fat” How to obtain this value.

7. Table 1. How to obtain these values.

8. Table 2. Move to Results and Discussion and explain the phenomenon.

9. 2.2. Microbiological Analysis. Give a reference.

10. 2.3. Biochemical Analysis. Give a more details.

Results and Discussion

11. 3.1 Microbiological Analysis. The English expression is too wordy and nonstandard. As the exhibited in the manuscript, the primary effect of the MAP on the quality properties of CCs in this work seems to be the inhibition of bacteria. More details should be presented, about the respective effects of nitrogen and hydrogen on bacteria in this MAP, as well as the reasons why choose these gas components and these gas concentrations.

12. “there was no significant difference between the sample groups after day 21”. Explain why and give references.

13. 3.2. Biochemical Analysis. More mechanism discussion should be given, about why the differences in treatment (different H2 concentration) result in the difference in biochemical results.

14. “The findings of CC sample biochemical analysis are shown in Table 2” Wrong table.

15. ... during the storage period of the CC samples (Table 1)” Wrong table.

16. “the acidic and lipolytic development in the control samples is related to microbial growth.” Give more details and several references.

17. 3.3. Color Characteristics. Give more mechanism about the changes in color properties of CC (Chemical change? Structure change?). Give several references.

18. 3.4. FA Composition of CC Samples and 3.5. FFAs of CC Samples. Obviously, there sections are not suitable to put here as their currently displayed form. It needs major adjustments in structure and content in the manuscript to make these parts appear appropriate.

Author Response

Dear reviewer.

I tried to make the revisions you requested as much as possible so that the study could be more original and increase the rate of scientific contribution. I hope I was successful. Thank you for your valuable contributions.

Edited and added parts are marked in yellow.

Below is the point-by-point reply.

Introduction

1: This method describes a new application technique of modified atmosphere packaging with nitrogen and hydrogen for foods with high-fat content.

2: Water buffalo milk is a nutritious alternative to cow's milk because it contains higher fat, protein, lactose and mineral content than cow's milk [1,2].

3: . The correlation between the size of the fat globules and the percentage of unsaturated fatty acids suggests that water buffalo milk butter has a longer shelf life than its bovine milk counterpart due to its higher percentage of solid fat. A higher percentage of solid fat would result in slower fat hydrolysis and less rancidity [3].

4: CC is produced in many Asian and Balkan countries under names such as kaymak, kajmak, kaimak or gemagh [4]. Although CC made from water buffalo milk tends to be preferred, other types of milk such as cow's, sheep's and goat's milk can be used for its production [4,5]. In the present study, we also preferred water buffalo milk for the preparation of CC.

5: Buffalo milk was obtained from local markets in Iğdır Turkey. A 50-liter buffalo milk was obtained from a mixture of the milk produced by 11 water buffaloes. The composition of raw milk was determined using a Milk Analyzer (Master Pro, Milkotester, Bulgaria) as presented in Table 1.

The CC samples were prepared in January 2021 in the laboratory of the Department of Food Engineering at IÄŸdır University. The raw milk was filtered, heated to 40±2 °C, and passed through a separator to obtain the cream. A manual household cream separator was used for seperation. The cream was analyzed for fat content by the Gerber method [6], and 68% fat was detected. The cream was pasteurized at 85 °C for 30 minutes and cooled down to 8-10 °C for 2 hours. The cream was then divided into 50-gram portions in a PET cups. The sample cups were stored at 4°C until the packaging procedure.

6: The cream was analyzed for fat content by the Gerber method [6].

7: The composition of raw milk was determined using a Milk Analyzer (Master Pro, Milkotester, Bulgaria)

8: Gas measurements taken after 28 days in the headspace of the packages are presented in Table 2. At the end of the storage period, the hydrogen content of the samples packaged with MAP1 decreased from 4% to 1.6%. It has been reported that the 0.1% hydrogen remaining in the package during storage still maintains its effectiveness as a reducing agent. [7]. A small amount of oxygen, which was thought to be trapped in the structure of the cream during the application, was found in the MAP-treated samples.

9: Samples from CC weighing 10 g were diluted with 90 mL Ringer's solution. Total aerobic mesophilic microorganisms and yeast molds were counted in triplicate on PCA (Merck, Germany) at 30 °C for 2 days and on potato dextrose agar (PDA, Merck, Germany) at 25 °C for 7 days, respectively. Colonies were counted and results are expressed as the logarithm of colony-forming units per gram (log CFU/g) [8].

10: CC samples were analyzed for titratable acidity as lactic acid (LA %) [9]. The lipolysis was determined as acidity value (ADV) by the BDI method [10]. 5 g of the sample was placed in butyrometers and 20 ml of BDI solution (prepared by dissolving 30 g of Triton X-100 and 70 g of sodium tetraphosphate in 1 l of distilled water) was added. The butyrometers were placed in boiling water and the oil was completely released. The samples were centrifuged again for 1 min and kept in a water bath at 57 °C for 5 min. A sufficient amount (1.5-2 g) of the oil collected in the upper part was taken and 5 ml of oil solvent (petroleum ether/n-propanol; 4:1 v/v) containing 0.1 g/l thymol blue indicator was added and titrated with 0.01 N tetra-n-butylammonium hydroxide.

The analyzes for the determination of the peroxide number were performed by spectrophotometry according to the AOAC standard technique [11]. For peroxide analysis, 0.3 g of the oil sample was removed and transferred to a 20 ml glass tube. 9.6 ml of chloroform/methanol (70:30 v/v) was added and the oil was dissolved by vortex mixing. Then, 0.05 ml of a 30% ammonium thiocyanate solution was added. Then, 0.05 ml of a ferric chloride solution (0.35% solution containing 2% 10 M HCl) was added. The blank was prepared using the same oil-free solutions as above, and the absorbance values of all tubes were determined by measuring at 500 nm. The concentration was determined from the curve prepared from the absorbance values of ferric chloride in a series (0.25-10 mg/L). The obtained peroxide values are expressed as meq O2/kg oil.

11: language checked.

The modifications in the air pressure inside the package caused by MAP affect the metabolism of the product and slow down the growth of microorganisms, which prolongs the shelf life of the product and maintains its quality and appearance [12]. By restricting O2, undesirable chemical and enzymatic reactions and microbiological development are inhibited in MAP systems. By lowering the oxidoreduction potential of the atmosphere to negative values using a reducing agent such as hydrogen gas, oxidation reactions are restricted because the abundance of free electrons available for biochemical reactions effectively neutralizes oxidative compounds such as the extremely harmful hydroxyl radical. The addition of molecular hydrogen selectively reduces oxygen radicals, limiting lipid oxidation (rancidity) in cells and the resulting proliferation of aerobic bacteria [13]. Many studies report that MAP applications inhibit microbial growth [12,14]. One study also reports that MAP with hydrogen application inhibits microbial growth [15].

12: It is probably due to the developing acidity in control samples slowed down the microbial growth. The increase in acidity and biochemical components that occur during the ripening of foods inhibit the growth of microorganisms [16].

13: Due to slower microbial growth, especially during the initial storage period in the MAP1 samples, acidic and lipolytic development were also slower. %100 nitrogen in MAP recommends for many foods. Nitrogen is an inert gas used to replace air in oxygen sensitive packaged products, and also is used to inhibit the growth of aerobic microorganisms [17]. But nitrogen and the reducing effect of hydrogen seem to slow the growth of microorganisms in the package even more.

14: cheched

15: cheched

16: The microflora of whole milk tends to be present in the cream component. Microorganisms associated with the lipid hydrolysis of triglycerides to free fatty acids can produce increased acidity, rancidity, and soapiness [17].

17: The light, process temperature, pH, or oxygen capacity can change the color pigments of foods [18]. Also spoilage by microorganism especially by yeast and molds can cause to surface discoloration [17].

18: section was checked and repositioned.

  1. Ménard, O.; Ahmad, S.; Rousseau, F.; Briard-Bion, V.; Gaucheron, F.; Lopez, C. Buffalo vs. cow milk fat globules: Size distribution, zeta-potential, compositions in total fatty acids and in polar lipids from the milk fat globule membrane. Food Chem. 2010, 120, 544–551.
  2. Abesinghe, A.; Vidanarachchi, J.K.; Islam, N.; Prakash, S.; Silva, K.; Bhandari, B.; Karim, M.A. Effects of ultrasonication on the physicochemical properties of milk fat globules of Bubalus bubalis (water buffalo) under processing conditions: A comparison with shear-homogenization. Innov. Food Sci. Emerg. Technol. 2020, 59, 102237.
  3. Romano, R.; Giordano, A.; Chianese, L.; Addeo, F.; Musso, S.S. Triacylglycerols, fatty acids and conjugated linoleic acids in Italian Mozzarella di Bufala Campana cheese. J. Food Compos. Anal. 2011, 24, 244–249.
  4. Cakmakci, S.; Hayaloglu, A.A. Evaluation of the chemical, microbiological and volatile aroma characteristics of Ispir Kaymak, a traditional Turkish dairy product. Int. J. dairy Technol. 2011, 64, 444–450.
  5. Akalin, A.S.; Tokusoglu, Ö.; Gönç, S.; Ökten, S. Detection of biologically active isomers of conjugated linoleic acid in kaymak. Grasas y Aceites 2005, 56, 298–302.
  6. James, C.S. Assessment of analytical methods and data BT - Analytical Chemistry of Foods. In; James, C.S., Ed.; Springer US: Boston, MA, 1995; pp. 5–12 ISBN 978-1-4615-2165-5.
  7. ALWAZEER, D. Reducing atmosphere packaging technique for extending the shelf-life of food products. J. Inst. Sci. Technol. 2019, 9, 2117–2123.
  8. Vidanagamage, S.A.; Pathiraje, P.; Perera, O. Effects of Cinnamon (Cinnamomum verum) extract on functional properties of butter. Procedia food Sci. 2016, 6, 136–142.
  9. Case, R.A.; Bradley Jr, R.L.; Williams, R.R. Chemical and physical methods. 1985.
  10. Jellema, A.; Anderson, M.; Heeschen, W.; Kuzdal-Savoie, S.; Needs, E.C.; Suhren, G.; Van Reusel, A. Determination of free fatty acids in milk and milk products. Bulletin of the International Dairy Federation no. 265 1991.
  11. Egan, H.; Kirk, R.S.; Sawyer, R. Pearsons Chemical Analysis of Foods. Churchill Livingstone. Edinburgh London, Melb. New York 1981, 591.
  12. Fernandes, L.; Pereira, E.L.; Fidalgo, M.D.C.; Gomes, A.; Ramalhosa, E. Effect of modified atmosphere, vacuum and polyethylene packaging on physicochemical and microbial quality of chestnuts (Castanea sativa) during storage. Int. J. Fruit Sci. 2020, 20, S785–S801.
  13. Sezer, Y.Ç.; Bulut, M.; Boran, G.; Alwazeer, D. The effects of hydrogen incorporation in modified atmosphere packaging on the formation of biogenic amines in cold stored rainbow trout and horse mackerel. J. Food Compos. Anal. 2022, 112, 104688.
  14. Kimbuathong, N.; Leelaphiwat, P.; Harnkarnsujarit, N. Inhibition of melanosis and microbial growth in Pacific white shrimp (Litopenaeus vannamei) using high CO2 modified atmosphere packaging. Food Chem. 2020, 312, 126114.
  15. Alwazeer, D.; Tan, K.; Örs, B. Reducing atmosphere packaging as a novel alternative technique for extending shelf life of fresh cheese. J. Food Sci. Technol. 2020, 1–11.
  16. Cotter, P.D.; Beresford, T.P. Microbiome changes during ripening. In Cheese; Elsevier, 2017; pp. 389–409.
  17. Batt, C.A.; Tortorello, M. Encyclopedia of food microbiology. 2 2014. List. List. monocytogenes.
  18. Kutlu, N.; Pandiselvam, R.; Kamiloglu, A.; Saka, I.; Sruthi, N.U.; Kothakota, A.; Socol, C.T.; Maerescu, C.M. Impact of ultrasonication applications on color profile of foods. Ultrason. Sonochem. 2022, 89, 106109.

Reviewer 2 Report

Introduction - no information about similar studies, methods - no clearly defined information, no clear information in tables, results - should be clarified and developed a clear discussion in the text, conclusions - no specific results, no structured information.

Author Response

Dear reviewer.

I tried to make the revisions you requested as much as possible so that the study could be more original and increase the rate of scientific contribution. I hope I was successful. Thank you for your valuable contributions.

Edited and added parts are marked in yellow.

Round 2

Reviewer 1 Report

Although many revisions have been performed, the manuscript still needs to be further improved for publication. Thus, one more major revision is given.

1. Water buffalo milk is a nutritious alternative to cow's milk. An unsolved problem is that water buffalo milk is also a kind of cows milk. Thus, it could be changed to conventional cow' milk.

2. Why did it use 4 % H2 in MAP1 as the only treated group while the other two groups are more likely as control groups? What if higher or lower H2 level? Give reasons and references.

3. 3.1 section. There is a lack of discussion on the variation in gas compositions, including why it happen and how it will affect the samples.

4. 2.3 section. l as unit should be changed to L.

5. 3.4 section. This change is not satisfactory. The change of pigment seems cannot explain the color change in milk, due to the fact that milk is milky white, and its main change during the storage is the increase of b* value, which meant that milk turns yellow.

6. 3.5 section. This change is not satisfactory. It still need a major adjustment in structure and content.

Reviewer 2 Report

Dear Authors,
Please correct the following changes:

Graphs overlap data are unclear. [there is only one figure 1, it is opaque, there is no clear description of this figure in the text, no statistical hypothesis]

Still no discussion [there is no discussion described at all before the conclusion section, this section should be included].

Conclusions - need to be grouped by specific results [the whole chapter should be improved, at the beginning group these conclusions according to the given task together with the result data, and expand these conclusions because there are too few of them]